# Pan-Genomic Insights into Rumen Microbiome-Mediated Short-Chain Fatty Acid Production and Regulation in Ruminants

**DOI:** 10.3390/microorganisms13061175

**Published:** 2025-05-22

**Authors:** Jingyi Shi, Hongren Su, Shichun He, Sifan Dai, Huaming Mao, Dongwang Wu

**Affiliations:** Yunnan Provincial Key Laboratory of Animal Nutrition and Feed, Faculty of Animal Science and Technology, Yunnan Agricultural University, Kunming 650201, China; shijingyi3@163.com (J.S.); shr2904317687@163.com (H.S.); heshichun0529@163.com (S.H.); 15987179618@163.com (S.D.); 1980539@ynau.edu.cn (H.M.)

**Keywords:** ruminants, rumen microbiome, short-chain fatty acids, pan-genomics, genetic diversity

## Abstract

The rumen microbiome represents a cornerstone of ruminant digestive physiology, orchestrating the anaerobic fermentation of plant biomass into short-chain fatty acids (SCFAs)—critical metabolites underpinning host energy metabolism, immune function, and environmental sustainability. This comprehensive review evaluates the transformative role of pan-genomics in deciphering the genetic and metabolic networks governing SCFA production in the rumen ecosystem. By integrating multi-omics datasets, pan-genomic approaches unveil unprecedented layers of microbial diversity, enabling precise identification of core functional genes and their dynamic contributions to carbohydrate degradation and SCFA biosynthesis. Notable advancements include the following: mechanistic insights into microbial community assembly and metabolic pathway regulation, highlighting strain-specific adaptations to dietary shifts; precision interventions for optimizing feed efficiency, such as rationally designing microbial consortia and screening novel feed additives through pan-genome association studies; and sustainability breakthroughs, demonstrating how targeted modulation of rumen fermentation can simultaneously enhance production efficiency and mitigate methane emissions. This synthesis underscores the potential of pan-genomics to revolutionize ruminant nutrition, offering a blueprint for developing next-generation strategies that reconcile agricultural productivity with environmental stewardship. The translational applications discussed herein position pan-genomics as a critical tool for advancing animal science and fostering a resilient livestock industry.

## 1. Introduction

Ruminant animals occupy a pivotal position in global agricultural systems, as they simultaneously deliver economic value and sustain ecological balance. Moreover, their unique digestive system enables efficient utilization of plant resources, yielding high-nutritional-value products and promoting sustainable development within agricultural ecosystems. The ruminant stomach comprises four compartments: the rumen, reticulum, omasum, and abomasum. The rumen is the most distinctive and crucial compartment, harboring a vast array of microorganisms (bacteria, protozoa, fungi, archaea, and phages) [1]. These microorganisms are capable of breaking down cellulose, hemicellulose, and other complex carbohydrates, converting them into volatile fatty acids, methane, and carbon dioxide. They also synthesize essential nutrients such as microbial protein, which are digested, absorbed, and broken down into amino acids by the host. Through gluconeogenesis, the tricarboxylic acid cycle, and direct oxidation, they provide energy for the host [2]. In addition, a large number of bacteriophages are also present in the rumen. By infecting bacteria and archaea, they play a crucial role in the rumen microbial ecosystem, mainly including aspects such as regulating the microbial community structure, influencing nutrient metabolism, and controlling methane emissions [3,4]. Consequently, the rumen microbial community plays a central role in feed fermentation and nutrient absorption, forming the foundation of ruminant digestive physiology [5,6,7].

When the rumen pH of ruminants (such as cattle and sheep) is stable (ranging from 6.0 to 7.0), the microbial community is healthy, the daily diet is mainly composed of fermentable carbohydrates (either fiber or starch), and the animals are in a normal physiological state (without metabolic diseases or stress). Short-chain fatty acids (SCFAs) are crucial products of rumen fermentation in ruminants, serving as a primary energy source, accounting for over 70% of the total energy requirements. The major SCFAs include acetate, propionate, and butyrate, with acetate and butyrate primarily utilized in lipogenesis [8]. Acetate, as a precursor for milk fat synthesis, directly influences the yield and quality of milk fat. For instance, modulating rumen fermentation to increase acetate production can enhance milk yield and milk fat content in dairy cows [9]. Butyrate exhibits anti-inflammatory and immunomodulatory effects, promoting epithelial cell proliferation and differentiation, strengthening the rumen barrier’s function, inhibiting the production of inflammatory factors, and reducing pathogen invasion, thereby improving the overall health of the animal [10]. Propionate is essential for maintaining blood glucose levels and energy balance; increased propionate production contributes to improved animal growth rate and feed conversion efficiency [8]. Furthermore, the generation and metabolism of SCFAs are closely linked to methane and nitrogen utilization and emissions. Therefore, when regulating rumen fermentation, it is possible to specifically reduce the production of acetic acid and butyric acid, thereby reducing methane emissions and effectively alleviating the greenhouse effect [11,12]. At the same time, by optimizing the composition of feed and using additives rationally, it is possible to improve the utilization efficiency of nitrogen, reduce nitrogen emissions, and mitigate environmental pollution problems. In conclusion, studying the generation, metabolism, and regulatory mechanisms of short-chain fatty acids is of great significance for improving the production performance and health status of ruminants, as well as reducing environmental pollution.

In recent years, pan-genomics has rapidly advanced, offering a more comprehensive perspective on biological system functionality by integrating multi-omics data, including genomics, transcriptomics, proteomics, and metabolomics. This technology enables in-depth investigation of microbial community complexity and functional diversity, constructing complete gene maps of microbial communities. It integrates genomic data from multiple strains or microbial communities to comprehensively reveal their genetic diversity and functional potential, and to analyze metabolic pathways and functional networks within microbial communities [13,14]. Preliminary studies in ruminant research have utilized pan-genomics to identify associations between key rumen microbial genes and increased short-chain fatty acid production, laying the groundwork for further research. Consequently, pan-genomic analysis aids in understanding the role of rumen microbes in feed fermentation and nutrient absorption, providing a scientific basis for optimizing feed formulations and additive use, thereby improving ruminant production performance and health [15]. Compared to traditional single-genome studies, it offers a more comprehensive analysis of microbial community genetic diversity and functional complexity [16].

Given this background, this review summarizes the applications and advancements of pan-genomics in the study of rumen microbes, short-chain fatty acid production, and regulation in ruminants. The aim is to comprehensively assess the potential of pan-genomics in optimizing ruminant production performance, health management, and environmental sustainability.

## 2. Rumen Microbiome and Pan-Genomics

The highly diverse microbial community in the rumen can convert the plant-based feed ingested by ruminants into absorbable nutrients such as volatile fatty acids and microbial proteins. Among them, bacteria account for more than 90%, mainly consisting of *Bacteroidetes* and *Firmicutes*, and they are the main force in the decomposition and fermentation of cellulose [17]. Archaea are mainly *Methanogens*. By utilizing hydrogen and carbon dioxide to produce methane, they maintain a low-hydrogen environment in the rumen. Although the number of protozoa is small, they play an important role in the decomposition of cellulose, the degradation of proteins, and the regulation of the bacterial community (by preying on bacteria) [18]. Although the content of rumen fungi is not high, they have unique advantages in the decomposition of lignocellulose [19]. Although the number of rumen fungi is small, they perform outstandingly in the decomposition of lignocellulose. Protozoa mainly assist in the degradation of cellulose through indirect methods such as the pretreatment of fibers and the regulation of the bacterial community. In comparison to the advantages of bacteria in terms of enzyme activity and direct degradation ability, their role is relatively limited. Bacteriophages regulate host bacteria through lytic and lysogenic cycles: they can directly lyse cellulose-decomposing bacteria and others, controlling the number of bacteria to maintain ecological balance. For example, they can reduce the number of methanogenic archaea. Additionally, they can integrate their own genomes to form prophages, enhancing the adaptability of host bacteria. Under environmental stress, prophages can be reactivated and enter the lytic state [3,4]. The various rumen microorganisms do not exist in isolation but exhibit intricate interrelationships. For instance, bacteria and fungi demonstrate synergistic activity in plant fiber degradation, where bacteria initiate the initial breakdown of cellulose, facilitating further degradation by fungi. Protozoal predation on bacteria, in turn, modulates bacterial populations, contributing to community stability. These interactions profoundly influence the rumen fermentation process and the production of short-chain fatty acids (Figure 1A).

### 2.1. Development and Application of Pan-Genomics

Pan-genomics, also referred to as integrative or pan-genomics, is a discipline dedicated to the study of the complete set of genes within a species. It aims to integrate, analyze, visualize, and annotate diverse omics data, including genomics, transcriptomics, proteomics, and metabolomics. By simultaneously analyzing data from various levels, pan-genomic approaches offer a more comprehensive perspective on the functional study of biological systems. The concept of pan-genomics was initially proposed by Tettelin et al. in 2005 during their research on *Streptococcus* [13], where they discovered that a single genome could not fully represent the genetic diversity of a species, thus introducing the concept of a pan-genome. With the advancement of high-throughput sequencing technologies, the scope of pan-genomics research has expanded to encompass more microorganisms, plants, animals, and even humans. It is utilized to study microbial adaptability, environmental and host interactions, and to elucidate intraspecies genetic diversity, with broad applications in breeding, disease research, and ecological studies [20]. In ruminant animal rumen microbiome research, pan-genomics is closely focused on the characteristics of rumen microbes and the regulation of short-chain fatty acid (SCFA) production (Figure 1B). For instance, through pan-genomic analysis of different rumen microbial strains, genes associated with key metabolic pathways for SCFA production can be precisely located, providing molecular targets for the targeted regulation of rumen fermentation. Furthermore, when studying the response of rumen microbes to different feed components, pan-genomics can be employed to decipher the intrinsic link between changes in microbial gene expression and fluctuations in SCFA yield.

Pan-genomics has broad and in-depth applications in microbial research, encompassing various fields such as genomic diversity and evolutionary analysis, pathogenic microorganisms, and host–microbe interactions. It provides essential tools for elucidating microbial gene diversity, functional characteristics, and evolutionary mechanisms. For instance, by comparing the genomes of different pathogenic strains, variable genes associated with pathogenicity can be identified, providing targets for vaccine development and anti-infective therapy. Furthermore, analyzing the pan-genome of pathogenic microorganisms can optimize strain performance, increase the yield of metabolites, or enhance environmental tolerance, leading to the development of more effective biocontrol strategies [21,22]. Finally, in host–microbe interaction studies, pan-genomics reveals the symbiotic mechanisms and immune responses between microbes and hosts by comparing the microbial genomes within different hosts.

### 2.2. Current Status of Pan-Genome Database Construction

Several rumen microbial genome databases have been developed and made publicly available, including the Rumen Microbial Genomics Network (RMGN), the Hungate1000 Project, the CAZy Database, and Integrated Microbial Genomes & Microbiomes [23]. These resources provide more comprehensive data for studying ruminant rumen microbes, facilitating significant advancements in research (Figure 1C). The construction of rumen microbial pan-genome databases involves data collection, assembly, and analysis. High-quality data collection is the initial step, primarily relying on metagenomic sequencing data from rumen content samples and genome data from isolated and cultured microorganisms [16,24]. Samples are typically sourced from ruminants of various breeds and under different feeding conditions to ensure data diversity and representativeness. With increasing sequencing demands and technological advancements, high-throughput sequencing technologies, such as Illumina, PacBio, and Oxford Nanopore, are widely used for sequencing rumen microbial genomes. Short-read sequencing (e.g., Illumina) offers high accuracy, while long-read sequencing (e.g., PacBio and Nanopore) aids in resolving complex genome region assembly issues [25].

Approaches to constructing pan-genomes encompass iterative assembly, multi-isolate de novo assembly, and graph-based assembly. For short-read data, tools such as MEGAHIT or SPAdes can be employed, while Canu or Flye are suitable for long-read data. Pan-genome analysis aims to elucidate the genetic diversity and functional attributes of the rumen microbiome [26]. Comparative genomics using tools like Roary or Panaroo can identify core genomes (genes shared by all strains) and dispensable genomes (genes unique to certain strains). Functional enrichment analysis of core and dispensable genomes can then be performed using EggNOG-mapper or InterProScan, with functional classification and pathway analysis conducted using GO and KEGG databases [27,28]. However, existing databases present certain limitations. Specifically, data completeness needs to be enhanced, as genomic data for some rumen microbes under specific environmental conditions or physiological states are lacking. Furthermore, variations in data formats and annotation standards across different databases pose challenges for data integration and comprehensive analysis. Future development of rumen microbial pan-genome databases should focus on expanding data collection to encompass a broader range of rumen microbes, including those from extreme conditions, while also standardizing data formats to construct more efficient, comprehensive, and user-friendly database platforms to meet growing research demands.

## 3. Analysis of Rumen Microbial Community Structure from a Pan-Genomics Perspective

The genetic composition and diversity of the rumen microbiome are fundamental to the digestive and metabolic functions of ruminants. Genomic data from rumen microorganisms are acquired through metagenomic sequencing or single-organism genome sequencing. These data undergo assembly and functional annotation, followed by comparative genomic analyses to assess functional diversity and evolutionary relationships [16,24]. Pan-genomic analyses reveal that the core genome of the rumen microbial community primarily encompasses genes involved in essential cellular functions and metabolic pathways, including glycolysis, the tricarboxylic acid cycle, amino acid synthesis and degradation, and DNA replication and repair. During different stages of rumen fermentation, various functional genes exhibit specific roles and interdependencies. For instance, in the initial stages of glycolysis, the expression products of hexokinase genes catalyze glucose phosphorylation, providing substrates for subsequent reactions. The expression of phosphofructokinase genes is regulated by the energy status within the rumen; high energy demands lead to increased expression of these genes, accelerating the glycolytic process. Furthermore, environmental adaptation genes, such as cellulase and hemicellulase genes, are upregulated in response to increased levels of cellulosic materials in the rumen [29]. The resulting degradation products, such as oligosaccharides, then enter basic metabolic pathways like glycolysis, providing energy for microbial growth and simultaneously stimulating the expression of genes associated with the tricarboxylic acid cycle, thereby promoting the efficient utilization of substrates by the microorganisms [25].

Pan-genomic approaches have elucidated the species and functional diversity of the rumen microbiome, which encompasses a diverse array of microorganisms, including bacteria, archaea, fungi, and protozoa. In 2013, Leahy et al. conducted a pan-genomic analysis of rumen archaea, revealing that the *Methanobrevibacter* genome is enriched with genes associated with methanogenesis [30]. This suggests that distinct microorganisms possess unique gene complements, enabling them to exert their functions through diverse metabolic pathways.

### 3.1. Impact of Various Factors on Rumen Microbiota Composition

The rumen microbiome plays a pivotal role in the digestion and nutritional status of ruminants. However, the structure of the rumen microbial community is dynamic, influenced by factors such as ruminant species, rearing environment, and diet composition. These factors induce alterations in environmental parameters like rumen pH, temperature, and nutrient concentrations, thereby affecting the growth and proliferation of rumen microbes (Figure 2). Growth refers to the increase in the volume or mass of cells or organisms, and its efficiency depends on the rate of anabolism. Proliferation, on the other hand, refers to the increase in the number of cells (through cell division), and its efficiency depends on the speed of the cell division cycle. In recent years, advancements in high-throughput sequencing technologies have facilitated in-depth investigations into the impact of various factors on rumen microbial community structure. Common research methodologies include 16S rRNA gene sequencing, which targets the highly conserved 16S rRNA gene to identify and classify bacteria in the rumen; metagenomic sequencing, capable of sequencing the entire microbial community DNA to explore the genetic diversity and functional potential; and metabolomics analysis [31], which focuses on the small-molecule metabolites to reflect the metabolic status of the rumen ecosystem.

The research conducted by Henderson et al. in 2015 and the research by Pitta et al. in 2016 have demonstrated significant disparities in the rumen microbial community structure among various ruminant species, such as cattle and sheep, potentially attributed to their distinct digestive physiology and dietary habits [19,32]. The rearing environment, encompassing factors like feeding management, stocking density, and hygiene protocols, also exerts an influence on the rumen microbial community structure [19]. For instance, *Prevotella* and fiber-degrading bacteria are dominant in the rumen of cattle, which is related to their requirement for high-fiber digestion. In contrast, in the rumens of sheep and goats, *Ruminococcus* and *Fibrobacter* are more abundant, which is associated with their adaptation to low-quality roughage. Compared with cattle raised in a confined feeding system, those under grazing conditions have a more diverse rumen microbial community structure. In contrast, the composition of the rumen microbial community in cattle raised in a confined feeding system is more stable [17,31]. Furthermore, the type of feed constitutes a pivotal factor affecting the rumen microbial community structure, as different feed types contain varying nutrient compositions, thereby impacting the growth and proliferation of rumen microbes. For example, a diet predominantly composed of roughage promotes the proliferation of cellulolytic bacteria, whereas a diet primarily consisting of concentrate feed favors the growth of amylolytic bacteria [33,34], Special diets, such as high-fat diets or those supplemented with tannins or plant extracts, selectively inhibit certain microbial populations, including protozoa and *Methanogens* [34]. Concurrently, the type of reproduction is another critical factor shaping the rumen microbial community structure. Naturally conceived offspring acquire maternal microbial communities through vertical transmission routes such as colostrum ingestion, coprophagy, and maternal licking, resulting in a microbiome composition closely resembling that of the dam [35]. In contrast, artificially inseminated or surrogate-born offspring exhibit rumen microbiota that are predominantly influenced by the surrogate dam or rearing environment rather than their genetic parents. This divergence arises due to disrupted microbial colonization patterns during early development [36]. These findings provide a theoretical basis for modulating the rumen microbial community structure to enhance the production performance and feed utilization efficiency of ruminants.

### 3.2. The Role of Microbial Genetic Variation

Genetic variation plays a crucial role in the ecological adaptation and functional differentiation of microorganisms. Microorganisms adapt to environmental changes through genetic variation and perform different functions in the ecosystem. Genetic variations in microorganisms mainly occur through means such as mutation, genetic recombination, and horizontal gene transfer (Figure 2). These variations enable microorganisms to rapidly adapt to environmental changes, such as temperature, pH, nutrients, etc. [37,38,39,40,41]. In 1991, the study by Lenski et al. showed that *Escherichia coli* adapted to a high-temperature environment through mutations during long-term experimental evolution [42]. From this, it can be known that mutation, as a basic form of genetic variation, can generate new alleles, enabling microorganisms to express specific genes under specific environmental conditions, thus adapting to environmental changes. In 2010, Davies’ research demonstrated that the spread of antibiotic resistance genes is achieved through horizontal gene transfer, which allows microorganisms to survive under antibiotic pressure [43]. However, the long-term use of antibiotics in rumen regulation will lead to the spread of antibiotic resistance genes [44]. Horizontal gene transfer is very common in microorganisms and can rapidly introduce new functional genes. For example, in rumen microorganisms, the butyric acid synthesis gene cluster is transferred from *Clostridium* to *Eubacterium* through a conjugative plasmid (such as pANH1) [45]. This horizontal gene transfer event enables *Eubacterium* to acquire new genetic material, significantly enhancing its butyric acid synthesis ability and potentially altering the overall SCFA production profile in the rumen. And the horizontal gene transfer (HGT) events of the succinate-propionate conversion genes (*fumC* and *mmdA*) enable the microbial community to maintain a high production of propionate (concentration > 30 mM) under low pH conditions [46]. Genetic variation not only affects the ecological adaptation of microorganisms but can also lead to functional differentiation. In 2008, Falkowski’s research showed that soil microorganisms decompose organic matter through different metabolic pathways, thus playing different roles in the carbon cycle, which indicates that genetic variation leads to the diversity of metabolic pathways [47].

When genetic variations occur in rumen microorganisms due to changes in environmental factors, the community structure can maintain relative stability within a certain period of time through the mutual regulation among microorganisms. Firstly, certain key species in the microbial community may quickly adapt to new environmental conditions through genetic variations, thus maintaining their ecological functions. For example, when the pH value drops, some microorganisms can acquire acid-resistant genes through horizontal gene transfer, enabling them to continue to survive and perform their functions in a low-pH environment [46]. Secondly, intricate interactions among species within the microbial community, such as competition, symbiosis, and predation, contribute to the regulation of community structure, preventing the overgrowth or extinction of any single species. For example, certain microbes may secrete metabolic products that inhibit the growth of other species, thus maintaining community equilibrium [43]. Furthermore, functional redundancy within the microbial community is a crucial factor, where different species may possess similar functions. When one species declines due to environmental changes, other species can occupy its ecological niche, thereby maintaining the overall function of the community. For example, in the rumen microbial community, multiple microbes have the ability to degrade cellulose; when one cellulolytic bacterium declines due to environmental changes, other cellulolytic bacteria can continue to perform this function, thus maintaining community stability [48].

In summary, pan-genomics provides a theoretical basis for modulating community structure, enhancing ruminant production performance, and improving feed utilization efficiency. Genes play a pivotal role in microbial ecological adaptation and functional differentiation via mechanisms such as mutation, recombination, and horizontal gene transfer. Although environmental fluctuations may induce genetic variations within the rumen microbiota, the community structure can maintain relative stability over time due to inter-microbial regulatory interactions and functional redundancy. This stability is crucial for sustaining ecosystem function and health.

## 4. Pan-Genomics and the Mechanism of Rumen Short-Chain Fatty Acid Production

Short-chain fatty acids (SCFAs) represent the primary end-products of ruminal fermentation in ruminants. SCFAs are primarily produced in the rumen via glycolytic and amino acid fermentation pathways. Rumen microorganisms degrade complex carbohydrates (e.g., cellulose, starch) and proteins into simpler substances such as monosaccharides (e.g., glucose) and amino acids [1,2]. These are then metabolized through glycolysis and amino acid fermentation to intermediate metabolites like pyruvate and acetyl-CoA, which are subsequently converted into SCFAs such as acetate, propionate, and butyrate [49,50] (Figure 3). Related to membrane transport: AE: Anion Exchanger, NHE: Na^+^-H^+^ Exchanger, MCT1: Monocarboxylate Transporter 1. Short-chain fatty acids and related substances: SCFA^−^: Short-Chain Fatty Acid anion, HSCFA: Unionized Short-Chain Fatty Acid.

### 4.1. Application of Pan-Genomics in the Mechanisms of Short-Chain Fatty Acid Production

In recent years, pan-genomics has gained widespread application in the study of rumen microbial communities. Stewart et al. conducted a pan-genomic analysis of 913 rumen microbial genomes, revealing that *Prevotella* spp. and *Butyrivibrio* spp. genomes are enriched with genes closely associated with short-chain fatty acid (SCFA) production. This discovery successfully identified key microbial species involved in SCFA production, including fibrolytic bacteria (e.g., *Fibrobacter succinogenes*, *Ruminococcus flavefaciens*, *Ruminococcus albus*), amylolytic bacteria (e.g., *Prevotella ruminicola*, *Streptococcus bovis*), acetogenic bacteria (e.g., *Prevotella* spp., *Clostridium* spp.), and butyrogenic bacteria (e.g., *Butyrivibrio fibrisolvens*, *Eubacterium rectale*) [24]. When the diet is predominantly roughage, the rumen pH is relatively high, and the temperature is suitable. Under these conditions, cellulase genes (e.g., the *celA* gene from *Fibrobacter succinogenes*) are induced by high pH and appropriate temperature. The promoter region binds with transcription factors, enhancing gene transcription levels and significantly increasing the expression of cellulases. This efficiently degrades cellulose to produce monosaccharides like glucose, providing ample substrate for SCFA production, thereby increasing the production of SCFAs such as acetate [51]. Conversely, when the diet is primarily concentrate, the starch content in the rumen increases, altering the nutrient concentration. The expression of amylase genes (e.g., the *amyA* gene from *Prevotella ruminicola*) is upregulated, breaking down starch to produce glucose. The expression of the pyruvate-ferredoxin oxidoreductase (*PFOR*) gene also changes, altering the direction of pyruvate metabolism and relatively increasing the proportion of propionate production [52].

In addition, with its unique advantage of integrating the genomic data of multiple microorganisms, pan-genomics can comprehensively and deeply reveal the genetic composition and functional diversity of the rumen microbial community. Take the Hungate1000 project as an example. This project conducted a combined analysis of single-cell genomics and metabolomics (scRNA-seq + scMetabolomics) on 1000 species of rumen microorganisms. Through scRNA-seq, RNA is extracted from single cells of rumen microorganisms, reverse-transcribed into cDNA, and then high-throughput sequencing technology is used to accurately obtain the gene expression information of individual microbial cells, revealing the functional states of different microorganisms in the rumen. scMetabolomics, on the other hand, understands the metabolic activity of microorganisms by separating, identifying, and quantitatively analyzing the metabolites of rumen microorganisms. The combined analysis of these two has significant advantages compared with the individual analysis. For example, scRNA-seq can detect that the expression of certain genes in a certain type of microorganism is upregulated under specific conditions, but it is unable to determine the actual metabolic effects of these gene expression products. However, scMetabolomics can detect the changes in the content of corresponding metabolites. The combination of the two can comprehensively and deeply analyze the metabolic networks and functional mechanisms of rumen microorganisms in different environments, providing more accurate and comprehensive data support for the study of the rumen microbial community. Furthermore, it reveals various types of microorganisms and functional genes related to the production of short-chain fatty acids [25]. Researchers, by means of pan-genomics analysis, have identified key functional genes involved in the production of short-chain fatty acids, such as cellulase genes, amylase genes, phosphotransacetylase (*pta*), acetate kinase (*ackA*), pyruvate-ferredoxin oxidoreductase (*PFOR*), butyrate kinase (*buk*), and phosphotransbutyrylase (*ptb*), etc. Among them, the core gene *ackA-pta* is highly conserved in all acetate-producing bacteria, and its expression level is positively correlated with acetate production [53,54,55]. These genes play a crucial role in the processes of cellulose degradation, starch decomposition, and the production of short-chain fatty acids.

### 4.2. The Synergistic and Competitive Relationships Among Different Microorganisms in the Process of Short-Chain Fatty Acid Production

The production of short-chain fatty acids (SCFAs) originates from a complex microbial fermentation process, in which multiple types of microorganisms are involved. These microorganisms exhibit both synergistic and competitive relationships with each other, collectively influencing the production efficiency and specific composition of SCFAs [56,57]. In the rumen of young ruminants, the microbial community has not yet fully matured, and there are significant differences in the synergistic and competitive relationships among microorganisms compared with those in adult ruminants. The oxygen content in the rumen of young ruminants is relatively high, and some facultative anaerobic bacteria dominate at this stage, showing weak synergistic effects with other microorganisms [31,58]. For example, in the early stage, *Enterobacteriaceae* bacteria in the rumen of young cattle mainly obtain energy through their own metabolic pathways, and their ability to cooperatively decompose substrates with fibrolytic bacteria and other microorganisms to produce SCFAs is limited. As ruminants grow and develop, the internal environment of the rumen gradually stabilizes, and the microbial community tends to mature, leading to an enhanced synergistic effect among microorganisms. In the rumen of adult ruminants, a stable synergistic relationship is formed between fibrolytic bacteria and acetate-producing bacteria. Fibrolytic bacteria decompose cellulose into monosaccharides, and acetate-producing bacteria rapidly utilize these monosaccharides to produce acetic acid, thereby improving the production efficiency of SCFAs, promoting the digestion and absorption of feed by ruminants, and meeting their growth and development needs. Meanwhile, the competitive relationships also change. Microorganisms in the rumen of adult ruminants compete more fiercely for substrates and ecological niches. For instance, the competition for hydrogen between *Methanogens* and acetate-producing bacteria is more pronounced in the adult stage, which affects the production ratio of SCFAs [31].

In terms of synergism, different microorganisms secrete unique enzymes, respectively, participating in different metabolic pathways. They jointly degrade complex carbohydrates and proteins, and convert substrates into short-chain fatty acids. In 2019, the study by Wallace et al. demonstrated that certain microorganisms in the rumen microbial community, through the integration of metabolic pathways, including *Prevotella ruminicola* and *Butyrivibrio fibrisolvens*, can significantly enhance the production efficiency of short-chain fatty acids. For example, after fibrolytic bacteria degrade cellulose into monosaccharides, acetate-producing bacteria and butyrate-producing bacteria further convert these monosaccharides into acetic acid and butyric acid. This synergistic effect not only increases the yield of short-chain fatty acids but also optimizes their composition ratio [59,60,61]. Meanwhile, microorganisms also create a suitable growth environment for each other by regulating environmental factors such as the rumen pH and redox potential.

In terms of competition, different microorganisms will compete to utilize the same substrates or metabolites. The study by Solden et al. has shown that there is a competitive relationship for hydrogen between *Methanogens* and acetate-producing bacteria in the rumen. In the rumen, cellulose is mainly decomposed by fiber-decomposing bacteria (such as *Ruminococcus* and *Fibrobacter*) to produce hydrogen, carbon dioxide, and short-chain fatty acids (such as acetic acid and propionic acid). Glycolytic bacteria (such as *Streptococcus bovis*) ferment soluble sugars to produce hydrogen and lactic acid. Protozoa release hydrogen through hydrogenase. *Methanogens* use hydrogen to produce methane, while acetate-producing bacteria use hydrogen to produce acetic acid. Acetogens can utilize hydrogen as an electron donor to reduce carbon dioxide or other one-carbon compounds (such as formic acid, methanol, etc.) into acetic acid. This process is known as acetogenesis and represents an important link in the carbon cycle in anaerobic environments [62]. This competitive relationship directly affects the production ratio of short-chain fatty acids. Especially under the condition of high-fiber feed, the activity of acetate-producing bacteria is enhanced, resulting in an increase in acetic acid production and a decrease in methane production [17,63]. In addition to this, microorganisms also compete for attachment to the surface of feed particles and for occupying spatial sites in the rumen. Pitta et al. studied the niche distribution of the rumen microbial community through metagenomic and spatially resolved metabolomic analyses. It was found that certain microorganisms can more effectively utilize substrates and produce short-chain fatty acids by occupying specific niches. For example, fibrolytic bacteria can more efficiently degrade cellulose and produce acetic acid by attaching to the surface of cellulose particles [32].

A thorough analysis of the synergistic and competitive relationships among different microorganisms helps us better understand the ecological mechanisms of the rumen microbial community and the regulatory mechanisms underlying the production of short-chain fatty acids. This, in turn, provides a solid theoretical basis for regulating the structure of the rumen microbial community, enhancing the production efficiency of short-chain fatty acids, and improving the production performance of ruminants.

## 5. Applications of Pan-Genomics in the Regulatory Strategies of Rumen Short-Chain Fatty Acids

Pan-genomics, with its powerful ability to analyze the diversity and functions of rumen microorganisms, plays a crucial role in the regulatory strategies of rumen short-chain fatty acids.

### 5.1. Applications of Pan-Genomics in Increasing the Yield and Proportion of Short-Chain Fatty Acids

In recent years, much progress has been made in the research of pan-genomics in regulating rumen short-chain fatty acids. By optimizing the structure of the microbial community and functional genes, and constructing efficient combinations of microbial communities, the yield and proportion of short-chain fatty acids can be effectively increased. Studies have shown that pan-genomic analysis can accurately analyze the gene functions of the rumen microbial community and optimize the production pathways of short-chain fatty acids.

Stewart et al., through analyzing the genomes of 913 rumen microorganisms, found that *Prevotella* spp. and *Butyrivibrio* spp. carried a high abundance of *ackA* and *buk* genes. Combined with the analysis using metabolomics and 16S rRNA gene sequencing technologies, these strains showed significant activity in the production of acetic acid and butyric acid. Through directional enrichment, the proportion of these strains in the rumen was increased by 15% [24], leading to an increase in the total output of short-chain fatty acids. In addition, through pan-genomic analysis, using functional gene markers (such as *pta*, *ackA*, *buk*, and *ptb*) or high-throughput screening technologies (such as microfluidic chips) can quickly identify and screen high-yielding strains. Combining pan-genomic data, Liu et al. developed a high-throughput screening method based on functional gene markers, and screened out *Fibrobacter succinogenes* (with high cellulose degradation ability) and *Prevotella ruminicola* (with high acetic acid production) from the rumen of water buffalos. After combining them, the acetic acid production increased by 12%, significantly improving the rumen fermentation efficiency [64].

In terms of microbial community regulation, pan-genomics can screen out key strains that efficiently produce short-chain fatty acids (SCFAs). By combining functional gene markers (such as *pta*, *ackA*, *buk*, and *ptb*) with high-throughput screening technologies (such as microfluidic chips), target strains can be rapidly identified and enriched. Research has found that using a directional enrichment strategy can increase the proportion of key SCFA-producing bacteria in the rumen by 15%, thereby optimizing the fermentation pattern and improving feed conversion efficiency. Regarding microbial modification strategies, Pankratz et al. used the CRISPR-Cas9 technology to perform gene editing on *Butyrivibrio fibrisolvens*, overexpressing the *buk* and *ptb* genes, which significantly increased the production of butyric acid [65]. Fu et al. introduced an exogenous succinate-propionate pathway into *Clostridium tyrobutyricum*. Through genomics verification of gene integration and metabolomics monitoring, the proportion of propionic acid increased from 15% to 28%, achieving directional regulation of the SCFA composition. The research also found that by knocking out the lactic acid production pathway (the *ldh* gene), the supply of acetyl-CoA can be further increased, thus improving the production efficiency of short-chain fatty acids [66,67]. Greening et al. found in their research that through the pan-genome, electron shunting genes (such as *hydA*) were discovered. By introducing an electron mediator (such as menaquinone) into the co-culture system, the production of propionic acid was promoted (via the succinate pathway), and the proportion of propionic acid increased from 20% to 45% [68]. Therefore, by constructing modular metabolic pathways, knocking out competitive pathways, or introducing exogenous pathways to modify the metabolic networks of microorganisms, the yield and proportion of short-chain fatty acids can be effectively increased (Figure 4).

In conclusion, these studies and examples demonstrate the application of microbial breeding and modification strategies based on pan-genomic data in increasing the yield and proportion of short-chain fatty acids, providing an important basis for optimizing the functions of the rumen microbial community.

### 5.2. Applications of Pan-Genomic Prediction Models in the Selection of Feed Additives and Precision Feeding Management

In recent years, as an emerging research method in genomics, pan-genomics can provide a more comprehensive understanding of the genetic diversity and functional genes of species, and has demonstrated great potential in the agricultural field [15]. Pan-genomics also has important applications in the optimization of feed additives. Firstly, by collecting genomic data of different varieties and strains of the target species, including reference genomes and re-sequencing data, and using bioinformatics tools to compare, assemble, and annotate these genomic data, the pan-genome of the species can be constructed. Subsequently, feeding data of the target species, including feed composition, additive use, growth performance, and health status, are collected. These data are then associated with the genomic information of individuals, and machine learning algorithms (such as Random Forest and Support Vector Machine) are employed to establish prediction models [69]. Mills et al. used the pan-genomic prediction model to study the effects of different feed additives on the rumen microbial community. The study found that certain additives (such as yeast extracts and plant extracts) can significantly enhance the activities of fibrolytic bacteria and acetate-producing bacteria, thus increasing the production of short-chain fatty acids. Through the precise selection of feed additives, researchers have successfully increased the yield of short-chain fatty acids by more than 10% [70]. The model can predict the responses of individuals with different genotypes to different feed additives. Then, according to the genotype information and growth stages of individuals, the most suitable feed additives can be selected, and the proportions of various nutrient components in the feed can be precisely regulated. Appropriate feed formulations can be developed to meet the nutritional needs of individuals, avoiding nutritional excess or deficiency, so as to improve feed utilization and animal production performance [70,71].

The prediction model can also be used to identify susceptible individuals. Pan-genomic information can be employed to screen out individuals with excellent traits, so that preventive measures can be taken in advance and directional breeding can be carried out to cultivate more superior varieties. Xiang et al. collected a large amount of genomic data of Large White pigs with different genotypes. They trained a model using genomic and metabolomic data, and conducted correlation analysis in combination with feeding data such as their growth performance and feed efficiency. Then, they established a prediction model by using machine learning algorithms. The research results showed that after selecting specific feed additives and adjusting the feed formula according to this model, the average feed conversion efficiency of Large White pigs increased by 12%, and the daily weight gain increased significantly [69]. This research achievement fully demonstrates the effectiveness of the pan-genomic prediction model in the selection of feed additives and precision feeding management (Figure 4).

### 5.3. Applications of the Combination of Genetic Selection and Pan-Genomics in Optimizing Rumen Fermentation Patterns

Rumen fermentation is a crucial part of the digestive process in ruminants, directly affecting feed utilization efficiency and animal production performance. Traditional genetic selection methods, through phenotypic selection and pedigree analysis, have achieved certain results in optimizing rumen fermentation patterns. However, there are still problems such as long cycles, low efficiency, and difficulties in analyzing complex genetic mechanisms. In recent years, the rise of pan-genomics has provided a new tool for the genetic breeding of ruminants. By integrating the genomic data of multiple individuals, it is possible to more comprehensively analyze the genetic variation and genetic diversity of the species. Through correlation analysis, functional genes related to rumen fermentation patterns can be mined, and more efficient and accurate molecular markers can be developed based on pan-genomic information for use in assisted selective breeding [71]. In terms of genetic selection, pan-genomics is helpful for screening ruminant individuals that can efficiently utilize short-chain fatty acids (SCFAs). Genomic–microbial interaction analysis reveals that different host genotypes may affect the structure of the rumen microbial community, thereby regulating the production of SCFAs. By combining the genome-wide association study (GWAS) with the pan-genomics method, host genes closely related to SCFA metabolism, such as SLC16A1 and ACSL1, can be screened out and used for breeding decisions.

Roehe et al. utilized pan-genomic data to study the relationship between the rumen microbial community and the host genome. The study found that certain host genes can significantly affect the structure and function of the rumen microbial community, thus influencing the production of short-chain fatty acids. Through genomic selection technology, researchers have successfully bred new varieties with excellent rumen fermentation performance [72]. Difford et al. identified host genes (such as DGAT1 and PLA2G2A) associated with the composition of rumen microorganisms through genome-wide association studies (GWASs). By combining with the metagenomic pan-genome data, they found that these genes regulated the fiber-degrading ability of microorganisms (such as the CAZymes gene cluster of *Ruminococcus*). In dairy cows carrying specific DGAT1 variants, the proportion of acetic acid in the rumen increased significantly [36]. By collecting the genomic data of ruminants of different varieties and strains, a reference pan-genome is constructed. Methods such as genome-wide association study (GWAS) are used to identify key genes and molecular markers related to rumen fermentation patterns. Then, through machine learning algorithms, a genomic selection model is developed to predict the rumen fermentation performance of individuals for early selection [60,64,71]. By combining traditional breeding methods with genomic selection technology, new varieties with excellent rumen fermentation performance can be bred to improve feed utilization efficiency and animal production performance (Figure 4).

In conclusion, the combination of genetic selection and pan-genomics provides new ideas and methods for optimizing rumen fermentation patterns. However, factors such as host genetics and microbial communities still need to be considered, and applying genomic selection technology to practical breeding also requires overcoming challenges such as costs and technology promotion. Nevertheless, with the continuous development of technology, this combination will play an increasingly important role in future ruminant breeding, providing strong support for the realization of efficient, environmentally friendly, and sustainable livestock industry development.

## 6. Challenges and Future Research Directions

### 6.1. Challenges

Pan-genomics holds great potential in the research field of short-chain fatty acids in the rumen of ruminants, providing a brand-new perspective for in-depth exploration of this field. However, currently, this research direction still faces many technical bottlenecks and limiting factors. The biological complexity and genetic diversity exhibited by the rumen microbial community are extraordinary. It encompasses a vast and diverse array of bacteria, archaea, fungi, and protozoa. Affected by multiple factors such as diet, health status, and individual differences, there are significant genomic differences among different individuals and species. This makes it extremely challenging to deal with a large number of variations and repetitive sequences during genome assembly, annotation, and pan-genomic analysis [19]. In terms of sequencing technology, although high-throughput sequencing technologies represented by Illumina can generate a large amount of short-read data with high accuracy, they have obvious shortcomings when assembling complex genomes, especially when dealing with repetitive and highly similar regions. In contrast, long-read sequencing technologies like PacBio and Oxford Nanopore can better resolve these complex regions, but they are costly and have limited throughput. Moreover, the high sequencing depth required to comprehensively cover various microorganisms further increases the complexity and cost of data processing. At the same time, pan-genomic analysis has a huge demand for computing resources such as high-performance computing clusters and large-capacity storage devices. Existing genome assembly and annotation algorithms have prominent limitations when dealing with complex microbial communities, especially in highly similar genomic regions. Furthermore, the functions of a large number of microbial genes have not been fully elucidated, which severely restricts a comprehensive understanding of the complex metabolic pathways of short-chain fatty acids. To analyze these interactions involving multiple metabolic pathways and microbial species, sophisticated experimental designs and advanced data analysis methods are urgently needed. In addition, to fully understand the mechanism of short-chain fatty acid production, it is necessary to integrate multi-omics data such as genomics, transcriptomics, proteomics, and metabolomics, which undoubtedly increases the difficulty of data analysis and interpretation [32,73].

Although pan-omics approaches have brought unprecedented insights to the study of short-chain fatty acids in the rumen of ruminants, it is crucial to overcome the above-mentioned challenges in order to fully unleash the potential of pan-genomics. With the continuous advancement of technology, overcoming these difficulties requires the collaborative efforts of multiple disciplines, the promotion of technological innovation, and the integration of various resources. Future research efforts should focus on developing more efficient sequencing technologies, improving existing data analysis algorithms, constructing more comprehensive functional annotation databases, and at the same time, making great efforts to strengthen the integration and sharing of multi-omics data. In this way, the research on short-chain fatty acids in the rumen of ruminants can be continuously pushed to new heights.

### 6.2. Future Research Directions

Although there have been some advancements in the research of rumen microorganisms in ruminants, due to technical limitations, there are still many challenges and new opportunities. Technologically, rumen microorganisms are complex. Although long-read sequencing technologies (such as PacBio and Oxford Nanopore) are beneficial for the assembly of complex genomes, they have high costs and limited throughput [74]. In the future, single-cell sequencing (scRNA-seq, scATAC-seq) can be combined with third-generation sequencing technologies to solve the problem of ambiguous gene attribution. However, it is necessary to optimize the process, reduce costs, and improve data quality. In terms of biomarker development, although functional genes, metabolites, and markers of the microbial community structure have potential, they face challenges such as specificity, stability, as well as technical and data interpretation issues. It is necessary to improve the integrated multi-omics analysis system [60]. Regarding the study of the interaction mechanism between microorganisms and the host, although there have been some achievements, in-depth analysis of the molecular mechanism and exploration of the co-evolutionary relationship face difficulties in terms of data and models. The time-series and spatially resolved multi-omics analysis, as well as the combination of single-cell metabolomics and spatial transcriptomics, also require overcoming technical and algorithmic challenges. For the regulation of the rumen microbial community, the precision nutrition strategy is difficult to be universally applicable due to individual and environmental differences, and gene editing technology faces ethical and safety issues [65]. Cross-species and cross-environment comparative studies are key points. Existing studies have a narrow scope and shallowly explore environmental factors. In the future, it is necessary to expand the scope and conduct in-depth research, but this faces challenges in terms of samples, data monitoring, and analysis [19]. In terms of functional annotation, deep learning tools can be helpful, but the data quality and model performance need to be improved. Relevant algorithms also need to be optimized to deal with data noise and interpretability issues.

In addition, due to the complexity of genomes, data integration, and limited computing resources, multi-disciplinary cooperation is required. The versatility and efficiency of the developed algorithms still need to be improved. There are issues regarding the update and quality control of open-source databases, and cloud computing platforms have potential data security risks [26]. In the future, it is necessary to improve the cooperation mechanism, unify data standards, and strengthen security protection to promote the development of research in this field.

At the technical integration level, spatial omics technologies (such as spatial transcriptomics and imaging mass spectrometry) can locate the distribution and metabolic activities of microorganisms in different regions of the rumen (such as the liquid phase and the solid-phase attachment layer). The integration of these technologies is expected to provide a more comprehensive understanding of the spatial heterogeneity of the rumen microbial community, which is crucial for optimizing rumen fermentation and improving ruminant production efficiency. For example, it can help identify the key regions where SCFA production is most active, guiding the development of more targeted interventions. For example, by combining fluorescence in situ hybridization (FISH) and nanoscale secondary ion mass spectrometry (NanoSIMS), the colonization dynamics of specific microorganisms (such as *Fibrobacter succinogenes*) on the surface of cellulose particles and the efficiency of acetic acid synthesis can be tracked, revealing the impact of the spatial heterogeneity of the microbial community on the local concentration of short-chain fatty acids and the absorption efficiency of the host [32]. Metabolic flux analysis, through the combination of stable isotope labeling (such as ^13^C-glucose, ^15^N-urea) and metabolic flux analysis (MFA), can quantify the real-time changes in the carbon and nitrogen metabolic fluxes in the rumen. For example, by using ^13^C labeling to track the distribution ratio of cellulose degradation products between acetic acid-producing bacteria and methane-producing bacteria, and combining with pan-genomic data to identify key regulatory genes (such as *ackA*, *mcrA*), it can provide targets for regulating methane production and increasing the yield of short-chain fatty acids [60]. By integrating the host genome-wide association study (GWAS) and microbial pan-genomic data, the role of host genetic variations (such as immune genes and metabolic regulatory genes) in shaping the microbial community structure can be explored [59,64,71]. For example, by comparing the host genomes and rumen microbial gene sets of different ruminant breeds (such as high-milk-fat cows and low-methane-emission sheep), co-evolutionary markers (such as the coordinated variation between host TLR genes and microbial LPS synthesis genes) can be identified, clarifying the impact of the host genetic background on the immune regulation of microbial functions [72].

Therefore, through technological innovation and the integration of multimodal technologies, there is hope in the future to break through the bottlenecks of data analysis and functional annotation, and to deeply reveal the co-evolutionary mechanism between the host and microorganisms. Single-cell sequencing, third-generation sequencing, and artificial intelligence will improve the accuracy and efficiency of genomic analysis. Spatial omics and metabolic flux analysis will analyze the ecological functions of microbial communities from the spatiotemporal dimension. Interdisciplinary collaboration will accelerate technological transformation, providing new strategies for precision feeding, low-carbon breeding, and host health management, and promoting the development of the livestock industry towards the direction of high efficiency, intelligence, and sustainability.

## 7. Conclusions

Pan-genomics has played an important role in the research on the production and regulation of short-chain fatty acids in the rumen of ruminants. By integrating multi-omics data, it has revealed the genetic diversity and functional potential of the microbial community, and has made significant contributions to optimizing the rumen fermentation pattern in combination with genetic selection and breeding, providing scientific support for the health, production efficiency, and environmental sustainability of ruminants. Future research should continue to focus on technological innovation, data integration, and interdisciplinary cooperation to overcome the current technical bottlenecks and promote the in-depth development of this field, ultimately contributing to the sustainable development of the livestock industry. Through pan-genomic technologies, a deeper understanding and optimization of the structure of the rumen microbial community can play an important role in improving the production performance of ruminants, ensuring animal health, and reducing environmental emissions. They have also promoted the integration of multi-omics data and technological innovation, laying a solid foundation for the realization of the sustainable development of the livestock industry.

## Figures and Tables

**Figure 1 microorganisms-13-01175-f001:**
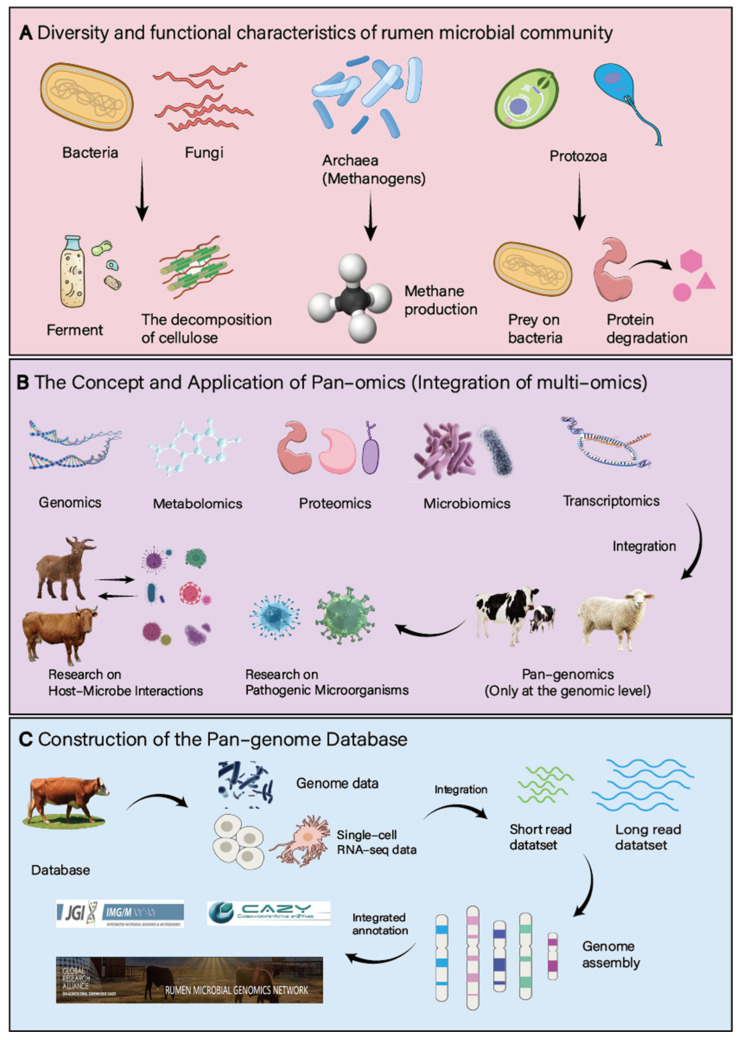
Rumen microbiome and pan-genomics. (**A**): Diversity and functional characteristics of rumen microbial community. (**B**): The concept and application of pan-genomics. (**C**): Construction of the pan-genome database.

**Figure 2 microorganisms-13-01175-f002:**
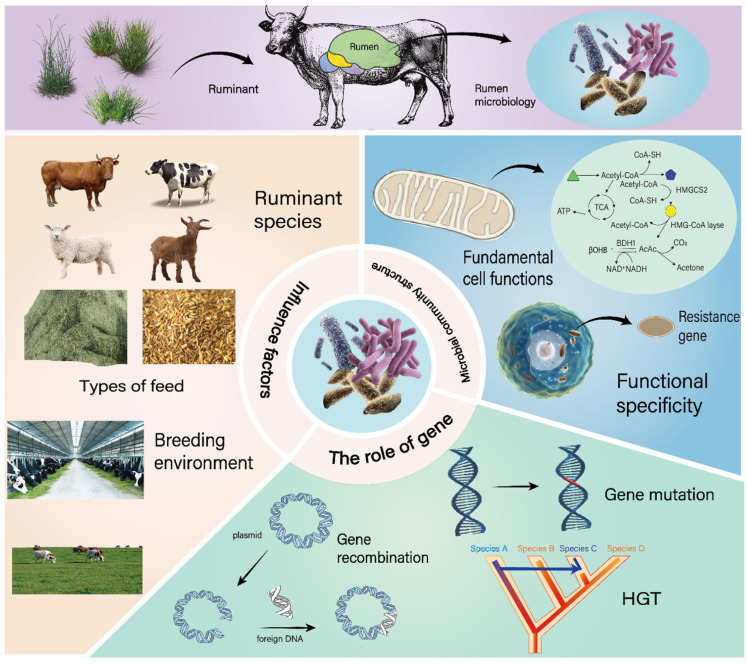
Rumen microbial community structure analysis from a pan-genomic perspective.

**Figure 3 microorganisms-13-01175-f003:**
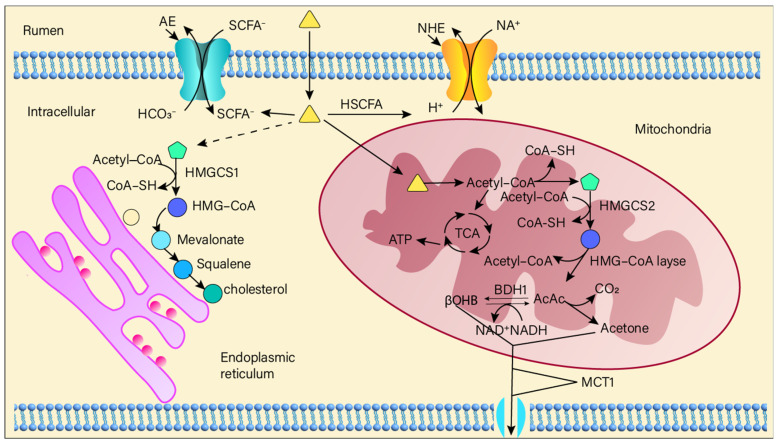
The generation mechanism of short-chain fatty acids when the microbial community in the rumen decomposes carbohydrates.

**Figure 4 microorganisms-13-01175-f004:**
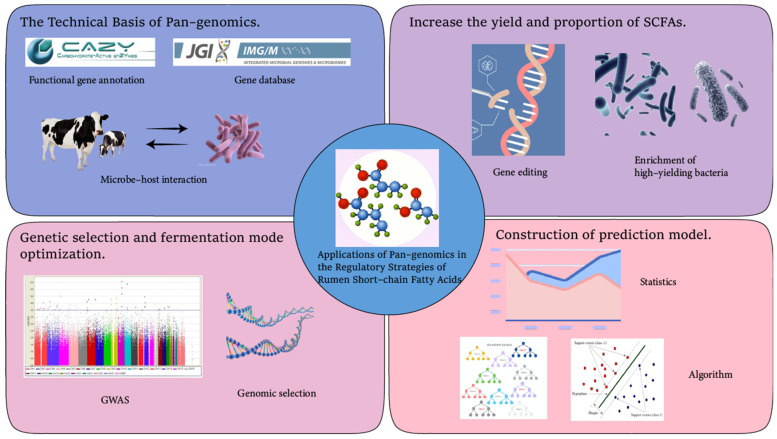
The application of pan-genomics in the regulation of SCFAs in the rumen.

## Data Availability

No new data were created or analyzed in this study.

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
