# Peer review of "Pan-Genomic Insights into Rumen Microbiome-Mediated Short-Chain Fatty Acid Production and Regulation in Ruminants"

_microorganisms, 2025, doi:10.3390/microorganisms13061175_

Round 1
Reviewer 1 Report
Comments and Suggestions for Authors
The paper offers a thorough analysis of pan-genomic insights into rumen microbiome-mediated SCFA synthesis and is extremely well-written and scientifically solid.
Strengths:
The article's structure is logical, the arguments are substantiated by recent references, the figures and tables are helpful, and the English used throughout the document is clear and fluid.
Minor Recommendations:
Despite the general great quality of the language, a little polishing could improve clarity even further by cutting down on minor repeats, especially in background passages.
To increase reader interest and flow, think about truncating a few longer paragraphs. In conclusion, I suggest making minor linguistic corrections but without making any significant changes. With just minor editorial adjustments, the text is of excellent quality and ready for publishing.
Comments on the Quality of English LanguageOutstanding overall:
The terminology is sophisticated, the wording is suitable for a scientific journal, and the academic English is good.
Sentences are coherent and well-formed, demonstrating good sentence structure. There were no significant grammatical mistakes or strange wording.
Correct scientific language: The technical terms are accurate and appropriately used.
Small points:
A few paragraphs repeat the same ideas, but this is not very noticeable and is typical in review articles. It could be further tightened with a little polishing if necessary. 9.5 out of 10 for language (extremely strong, nearly publication-ready).
Author Response
We feel great thanks for your professional review work on our article. As you are concerned, there are several problems that need to be addressed. According to your nice suggestions, we have made extensive corrections to our previous draft, the detailed corrections are listed below. The reviewer comments are laid out below in italicized font and specific concerns have been numbered. Our response is given in normal font and changes/additions to the manuscript are given in the red text.
Comments 1: Despite the general great quality of the language, a little polishing could improve clarity even further by cutting down on minor repeats, especially in background passages.
Response 1: We think this is an excellent suggestion, and we tried our best to improve the manuscript and made some changes to the manuscript. These changes will not influence the content and framework of the paper.
Line 68-83, Line 106-139, Line 224-227, Line 353-355, Line 368-370, Line 509-511. We have deleted the minor repetitions and polished the language content in this paragraph.
Comments 2: To increase reader interest and flow, think about truncating a few longer paragraphs. In conclusion, I suggest making minor linguistic corrections but without making any significant changes.
Response 2: We think this is an excellent suggestion.
Line 106-119, Line 224-227, Line 353-355, Line 368-370, Line 509-511. We have deleted some repetitive language and paragraphs to shorten the text.
Comments 3: A few paragraphs repeat the same ideas, but this is not very noticeable and is typical in review articles.
Response 3: Thanks for your suggestion. We have tried our best to polish the language in the revised manuscript and reduced the repetitions.

Reviewer 2 Report
Comments and Suggestions for Authors
The manuscript generally requires format modifications, is too long, and is sometimes repetitive. The use of abbreviations requires checking, and several statements require support.
About the content, the authors excluded Phages from the rumen microbiome. Pan-genomic approaches did not consider the phages? Please, explain to the reader in the manuscript.
L37: What about phages?
L40-41: Please rewrite. Microbial protein providing energy?
L44-45: Support it, and please be more specific about 70% for each ruminant in any conditions (diet, environment, species, etc)?
L47-49: Please, support it.
L53-55: Please, support it.
L55-63: Repetitive and confusing, please rewrite.
L85-86: Repetitive, it was mentioned in the introduction with the same words.
L97-98: Phages need to be considered.
L103: Figure 1 has several sections, A, B, etc. Please indicate to the reader which sections to pay attention to.
L120: Figure 1 has several sections, A, B, etc. Please indicate to the reader which sections to pay attention to.
L143: Please indicate to the reader which sections to pay attention to.
L178-180: Repetitive, it was mentioned many times before.
L191-195: support it
L212: What are the differences between growth and proliferation? What about efficiency ?
L221-223: Please cited the researchs
L226-227: Support it
L235: by microbial or by host?
L246-248 It was in rumen microbes? Currently, there are many concerns about using antibiotics to modulate rumen fermentation because antibiotic resistance answer in this microbiome. The authors are encouraged to provide the readers know more information about it
L281-287: Rewrite to be more concise and avoid being repetitive with several sentences mentioned before.
L295: Repetitive
L305: Where? by whom?
L315-327: Support it
L365-367: Support it
396-402: all acetate-producing bacteria? Please be more specific on the pathway to make this happen. Until my know, the main pathway to acetate production in the rumen is not associated with the consumption of H.
L416: This section can be reduced, without 3 subsections, because there are few studies reported in each one. Thus, instead of having 3 conclusions for each subsection, authors can write just one, and be more concise to the reader.
L421-422: Repetitive, it can be deleted
L429-431: The authors cited just four studies, which means that the application is still too low. Authors should be considered it during the discussion
Author Response
We feel great thanks for your professional review work on our article. As you are concerned, there are several problems that need to be addressed. According to your nice suggestions, we have made extensive corrections to our previous draft, the detailed corrections are listed below. The reviewer comments are laid out below in italicized font and specific concerns have been numbered. Our response is given in normal font and changes/additions to the manuscript are given in the red text.
Comments 1: The manuscript generally requires format modifications, is too long, and is sometimes repetitive. The use of abbreviations requires checking, and several statements require support.
Response 1: We think this is an excellent suggestion. We have revised the format, reduced the length, and removed the repetitions. We have also checked and explained the use of acronyms, and supplemented statements in the text with supporting information.
Line 68-83, Line 106-139, Line 224-227, Line 353-355, Line 368-370, Line 509-511. We have deleted some repetitive language and paragraphs to shorten the text.
Line 376-379. We clarified the meanings of all the acronyms in the text and arranged them by category.
Comments 2: About the content, the authors excluded Phages from the rumen microbiome. Pan-genomic approaches did not consider the phages? Please, explain to the reader in the manuscript.
L37: What about phages?
Response 2: We think this is an excellent suggestion.
Line 37, Line 44-48. We considered and added bacteriophages in rumen microorganisms and their functions, and provided explanations in the following text, and provided references for support.
Comments 3: L40-41: Please rewrite. Microbial protein providing energy?
Response 3: Thanks for your suggestion.
Line 42-44. We rewrote this part and explained the process by which microbial proteins are digested and absorbed by the host, after which they provide energy.
Comments 4: L44-45: Support it, and please be more specific about 70% for each ruminant in any conditions (diet, environment, species, etc)?
Response 4: Thanks for your suggestion.
Line 51-54. We have provided more specific descriptions of the situations under certain dietary, environmental, species and other conditions.
Comments 5: L47-49: Please, support it.
Response 5: Thanks for your suggestion.
Line 60. We have supplemented the references [9] to support the content of the text.
Comments 6: L53-55: Please, support it.
Response 6: Thanks for your suggestion.
Line 66. We have supplemented the references [8] to support the content of the text.
Comments 7: L55-63: Repetitive and confusing, please rewrite.
Response 7: Thanks for your suggestion.
Line 68-83. We have deleted the repetitive sentences and re-written this part according to the Reviewer's suggestion.
Comments 8: L85-86: Repetitive, it was mentioned in the introduction with the same words.
Response 8: Thanks for your suggestion.
Line 106-119. We deleted the repetitive sentences and reorganized the language.
Comments 9: L97-98: Phages need to be considered.
Response 9: Thanks for your suggestion.
Line 134-139. We have considered the roles of bacteriophages and provided references for support.
Comments 10: L103: Figure 1 has several sections, A, B, etc. Please indicate to the reader which sections to pay attention to.
Response 10: Thanks for your suggestion.
Line 145. We indicated the readers to pay attention to the Section A of Figure 1.
Comments 11: L120: Figure 1 has several sections, A, B, etc. Please indicate to the reader which sections to pay attention to.
Response 11: Thanks for your suggestion.
Line 162. We indicated the readers to pay attention to the Section B of Figure 1.
Comments 12: L143: Please indicate to the reader which sections to pay attention to.
Response 12: Thanks for your suggestion.
Line 188. We indicated the readers to pay attention to the Section C of Figure 1.
Comments 13: L178-180: Repetitive, it was mentioned many times before.
Response 13: Thanks for your suggestion.
Line 224-227. We deleted the repetitive sentences.
Comments 14:L191-195: support it
Response 14: Thanks for your suggestion.
Line 242. We updated and provided references to support the content.
Comments 15: L212: What are the differences between growth and proliferation? What about efficiency ?
Response 15: Thanks for your suggestion.
Line 260-263. We clarified the distinction between growth and proliferation, and differentiated their efficiencies.
Comments 16: L221-223: Please cited the researchs
Response 16: Thanks for your suggestion.
Line 272-273. We cited the research of Henderson et al.
Comments 17: L226-227: Support it
Response 17: Thanks for your suggestion.
Line 275. We updated the references for support.
Comments 18: L235: by microbial or by host?
Response 18: Thanks for your suggestion.
Line 303. We have revised the title, which is The Role of Microbial Genetic Variation.
Comments 19: L246-248 It was in rumen microbes? Currently, there are many concerns about using antibiotics to modulate rumen fermentation because antibiotic resistance answer in this microbiome. The authors are encouraged to provide the readers know more information about it
Response 19: Thanks for your suggestion.
Line 317-318. The relevant research by Davies was not conducted on rumen microorganisms. Currently, research on the regulation of rumen fermentation by antibiotics still faces bottlenecks. We have made every effort to provide relevant content and references.
Comments 20: L281-287: Rewrite to be more concise and avoid being repetitive with several sentences mentioned before.
Response 20: Thanks for your suggestion.
Line 353-355. We deleted the redundant parts within the sentences and rewrote and simplified the structure.
Comments 21: L295: Repetitive
Response 21: Thanks for your suggestion.
Line 368-370. We deleted the redundant parts within the sentences.
Comments 22: L305: Where? by whom?
Response 22: Thanks for your suggestion.
Line 381-382. We have revised the picture title to The generation mechanism of short-chain fatty acids when the microbial community in the rumen decomposes carbohydrates.
Comments 23: L315-327: Support it
Response 23: Thanks for your suggestion.
Line 399. We have updated the references and provided support for the text content.
Comments 24: L365-367: Support it
Response 24: Thanks for your suggestion.
Line 405. We have updated the references and provided support for the text content.
Comments 25: 396-402: all acetate-producing bacteria? Please be more specific on the pathway to make this happen. Until my know, the main pathway to acetate production in the rumen is not associated with the consumption of H.
Response 25: Thanks for your suggestion.
Line 479-483. Line 485-488. We have re-explained the specific processes of hydrogen production and the utilization of hydrogen by acetogens to produce acetic acid, and provided references for support.
Comments 26: L416: This section can be reduced, without 3 subsections, because there are few studies reported in each one. Thus, instead of having 3 conclusions for each subsection, authors can write just one, and be more concise to the reader.
Response 26: Thanks for your suggestion
We have reconsidered this part. It is difficult to express this content by combining the three subsections. We have added more research and references to support the content of the article, making this part more substantial and enriching.
Comments 27: L421-422: Repetitive, it can be deleted
Response 27: Thanks for your suggestion.
Line 509-511. We deleted the redundant parts within the sentences.
Comments 28: L429-431: The authors cited just four studies, which means that the application is still too low. Authors should be considered it during the discussion
Response 28: Thanks for your suggestion.
Line 552-556, Line 628-634. We have cited more research and provided references for support, making this part of the content richer and more substantial.
Reviewer 3 Report
Comments and Suggestions for Authors
This review analyzes a very interesting topic: the pangenomic study of the ruminal microbiota. Its perspective is original and lays the groundwork for future studies. Its publication could be very interesting but requires some changes, which are described below. Major aspects: It would be important for the review to highlight the differences in ruminal pangenomics based on different factors. The differences between ruminant species are not explored in depth (very few cases mention whether they refer to cattle, sheep, goats, and other animals), nor between diets, nor between other factors such as type of breeding (only age differences are highlighted). These aspects would enrich this review and must be addressed for the work to be published. Minor points: Line 94: Clarify that the role of portozoans in cellulose degradation is less than that of bacteria. Figure 1 does not clearly differentiate between pangenomics and pan-omics. Figure 3 should clarify the acronyms (regardless of whether they are stated in the text). It would be important to include glycolysis and amino acid degradation in this same figure, as well as pyruvate, which is mentioned in the text.
Author Response
We sincerely thank the editor and all the reviewers for their valuable feedback, which we used to improve the quality of the manuscript. The reviewers' comments are listed in italic font, and the specific questions are numbered. Our responses are given in normal font, and the changes/additions to the manuscript are given in red text.
Comment 1: It is very important that this review highlights the differences in rumen pangenomics based on different factors. Differences between ruminant species are not explored in depth (a few cases are mentioned whether they refer to cattle, sheep, goats, and other animals), nor are differences between diets, nor differences between other factors, such as reproductive type, investigated (only age differences are emphasized). These aspects would enrich this review and must be addressed for the work to be published.
Response 1: We think this is a good suggestion. We have rewritten this section based on the reviewers' suggestions.
Lines 278-284, 290-300. We have added an in-depth discussion of differences between ruminant species, diets, and other factors (such as reproductive type), and attached references.
Comment 2: Line 94: Clarify that protozoa play a smaller role than bacteria in cellulose degradation.
Response 2: We think this is a good suggestion. We have rewritten this section based on the reviewer's suggestion.
Lines 129-134. We have clarified that protozoa play a significantly smaller role than bacteria in cellulose degradation and have attached references.
Comment 3: Figure 1 does not clearly distinguish between pan-genomics and pan-omics.
Response 3: We think this is a good suggestion. Figure 1. We have revised the content of the figure and made a more detailed distinction between pan-genomics and pan-omics.
Comment 4: Figure 3 should clarify the acronyms (regardless of whether they are stated in the text). It is important to include glycolysis and amino acid degradation in the same figure, as well as the mention of pyruvate in the text.
Response 4: We think this is a good suggestion.
Lines 376-379. We have clarified the meaning of all acronyms in the text and arranged them by category.

Round 2
Reviewer 2 Report
Comments and Suggestions for Authors
The authors have considered my suggestions and the manuscript has been improved
Reviewer 3 Report
Comments and Suggestions for Authors
In the present form the manuscript can be accepted